# Value of Somatostatin Receptor Scintigraphy with ^99m^Tc-HYNIC-TOC in Patients with Primary Sjögren Syndrome

**DOI:** 10.3390/jcm8060763

**Published:** 2019-05-30

**Authors:** Luz Kelly Anzola, Josè Nelson Rivera, Rudi A. Dierckx, Chiara Lauri, Stefano Valabrega, Filippo Galli, Sergio Moreno Lopez, Andor W. J. M. Glaudemans, Alberto Signore

**Affiliations:** 1Nuclear Medicine Unit, Department of Medical-Surgical Sciences and of Translational Medicine, Faculty of Medicine and Psychology, “Sapienza” University, 00161 Rome, Italy; chialau84@hotmail.it (C.L.); filippo.galli@hotmail.com (F.G.); alberto.signore@uniroma1.it (A.S.); 2Nuclear Medicine Unit, Clinica Colsanitas, 11021 Bogotà, Colombia; 3Medical Imaging Center, Department of Nuclear Medicine and Molecular Imaging, University of Groningen, University Medical Center Groningen, 9700 Groningen, The Netherlands; r.a.dierckx@umcg.nl (R.A.D.); a.w.j.m.glaudemans@umcg.nl (A.W.J.M.G.); 4Internal Medicine Unit, Clinica Colsanitas, 11021 Bogotà, Colombia; jonerimo@yahoo.com; 5Surgery Unit, Department of Medical-Surgical Sciences and of Translational Medicine, Faculty of Medicine and Psychology, “Sapienza” University, 00161 Rome, Italy; stefano.valbrega@uniroma.it; 6Epidemiology Department, National University of Colombia, 11021 Bogotà, Colombia; smmorenol@unal.edu.co

**Keywords:** Sjögren syndrome, ^99m^Tc-HYNIC-TOC, somatostatin receptor scintigraphy, inflammation, salivary glands

## Abstract

Objectives: Primary Sjögren syndrome (SS) is diagnosed based on the American European Consensus Group (AECG) criteria, but lacks specificity, not only in the involvement of salivary glands, but also in extra-glandular involvement. Whole-body somatostatin receptor scintigraphy with ^99m^Tc-HYNIC-TOC scintigraphy could overcome these limitations. The aims of this study were to evaluate salivary gland uptake of ^99m^Tc-HYNIC-TOC in untreated patients with de-novo diagnosis of SS as compared to control subjects and as compared to conventional sialoscintigraphy with ^99m^TcO_4_^−^. We also aimed to evaluate the involvement of joints. Methods: ^99m^Tc-HYNIC-TOC was used with SS patients and uptake in joints and salivary glands was analyzed semi-quantitatively. Patients also underwent ^99m^TcO_4_ sialoscintigraphy. The control group that we analyzed consisted of 30 patients with neuroendocrine tumors. Results: Fifty-two females and 10 males fully met the AECG criteria for SS, and were included. A target background ratio (TBR) >1.18 in submandibular glands correctly classified 93% of the patients with SS in comparison to 27% for ^99m^TcO_4_ sialoscintigraphy. The area under the curve (ROC) analysis for TBR in submandibular glands was 0.95. In joints there was a huge variety in uptake. The median TBR was significantly higher in salivary glands in patients with SS compared to controls. Conclusions: ^99m^Tc-HYNIC-TOC scintigraphy identified active inflammatory processes not only in the salivary glands, but, unexpectedly, also in many joints in patients with primary SS, contrary to popular belief. This technique provides an objective parameter to evaluate the inflammation burden in salivary glands and joints and could be used to evaluate response to treatment.

## 1. Introduction

Sjögren syndrome (SS) is a systemic autoimmune disease that primarily affects the salivary and lacrimal glands. It usually causes a persistent dryness of the mouth and eyes due to lymphocytic infiltration and impairment of the exocrine glands [1,2]. The presence of circulating auto-antibodies that evoke an autoimmune response by cytokines derived from both T and B cell lymphocytes are thought to contribute to the inflammation and destruction of the glandular tissue [3]. Primary SS is characterized only by the presence of these exocrinopathies, whereas in secondary Sjögren these disorders are associated with other autoimmune diseases [1].

Primary SS has a prevalence of about 0.5% in the general population, with a female predominance of 9:1, which is approximately similar to that of systemic lupus erythematosus (SLE) [4]. SS is commonly included in the spectrum of connective tissue diseases and sometimes shows multisystemic involvement with a large range of clinical and serological manifestations. Besides the disease-specific exocrine manifestations, SS may be characterized by the involvement of the joints, skin, lung, kidneys, and nervous system, and it is associated with the production of a variety of autoantibodies [5]. The American–European Consensus Group (AECG) criteria, published in 2002 [6], was adopted as the gold standard criteria in Europe and in the United States to diagnose SS. For primary SS, the presence of four out of six items showed good sensitivity (93.5%) and specificity (94%) [7,8]. The ACR-EULAR initiative has decided to reunite the criteria to make clinical studies and therapeutic trials comparable [9]. In this new approach, sialoscintigraphy is not included as diagnostic criterion. As a matter of fact, although sialoscintigraphy was considered part of the diagnostic criteria for SS for AECG, this is a technique that lacks specificity and is not commonly used anymore [10]. Diagnosing secondary SS has not yet been addressed by the AECG, however, in practice it is usually required to fulfill the criteria for primary SS and to additionally fulfill the American College of Rheumatology (ACR) criteria for an established connective tissue disease such as rheumatoid arthritis (RA), SLE, dermatomyositis, myositis, or biliary cirrhosis [11]. Particularly in the clinical diagnostic setting where SS patients present with severe dryness, positive autoantibodies, and positive lip biopsies, it is important to assess the extent of extra glandular involvement for therapy decision making. Therefore, a standardized whole body imaging technique to determine other sites of disease manifestations is highly needed. Somatostatin is a hormone that regulates several physiological cell processes via specific receptors expressed throughout the body, particularly by nerve cells, many neuroendocrine cells, and cells mediating inflammation and the immune response [12]. Its physiological actions are initiated by binding to G-protein-coupled somatostatin receptors (SSTR1–SSTR5) [13]. High expression levels of SSTRs have been observed in tumor cells as well as neo-angiogenic and peri-tumoral vessels, epithelioid cells, proliferating synovial vessels, and activated lymphocytes and monocytes [14]. Besides overexpression in several autoimmune and granulomatous diseases, such as RA, SLE, Schönlein–Henoch, autoimmune uveitis, ulcerative colitis, sarcoidosis, tuberculosis, and Crohn’s disease, SSTR overexpression is also well known in patients with SS [15,16].

A strong interest in SSTRs as targets for in vivo diagnostic and therapeutic purposes followed the availability of somatostatin analogues [17]. Several molecules that bind to SSTR2 and SSTR5 receptors isoforms, and with lower affinity to SSTR3 [18], have been labelled with ^111^Indium (^111^In-octreotide^19^ and ^111^In-DTPA-D-Phe(1)-octreotide (OctreoScan^®^, Mallinckrodt). In order to overcome some limitations in the use of OctreoScan^®^, including the high costs and suboptimal physical features of ^111^In, somatostatin analogues have also been labelled with ^99m^Technetium, for example using depreotide [15]^15^ and ^99m^Tc–EDDA/Tricine-HYNIC-Tyr(3)-Octreotide (^99m^Tc-HYNIC-TOC) [19]. The latter has recently been used in clinical settings including neuroendocrine tumors (NET) and a number of chronic inflammatory diseases [20,21,22], where uptake of the tracer was described not only in the main compromised organs, but also in the salivary glands. ^99m^Tc-HYNIC-TOC has a high affinity for SSTR2, SSTR3, and SSTR5, and has demonstrated potential utility in the diagnostic work-up and treatment evaluation of chronic inflammatory diseases [22]. Although the use of ^99m^Tc-HYNIC-TOC has been extensively described for malignancies [15] and chronic inflammatory processes including secondary SS [7,17], the diagnostic, prognostic, and therapeutic potential in primary SS has, to our knowledge, not been previously addressed. 

The main objective of this study was to evaluate the characteristics of ^99m^TcHYNIC-TOC distribution in the salivary glands of patients with newly diagnosed SS based on a semi-quantitative analysis. The secondary objectives were (1) to correlate our findings in the salivary glands to the conventional sialoscintigraphy with ^99m^TcO_4_^−^, (2) to evaluate extra-glandular involvement of the joints, and (3) to compare the findings in salivary glands and joints with control patients without SS that underwent ^99m^Tc-HYNIC-TOC scintigraphy. 

## 2. Experimental Section

### 2.1. Materials and Methods

#### 2.1.1. Study Design

We retrospectively analyzed a consecutive cohort of 62 patients with de novo diagnosis of primary SS who underwent ^99m^Tc-HYNIC-TOC scintigraphy at the nuclear medicine unit of Clinica Colsanitas in Bogotá between January 2013 and November 2016. Furthermore, regarding the negative control group, we evaluated the uptake of salivary glands uptake and joints uptake of ^99m^Tc-HYNIC-TOC in 30 patients in whom the scan was performed for staging of NET. In order to avoid an influence on salivary gland uptake in this control group population, only subjects with negative scans or with a very low tumor burden (located only in the abdominal area) were chosen.

#### 2.1.2. Radiopharmaceutical Details

^99m^Tc-HYNIC-TOC was prepared in the radiopharmaceutical department from a commercially available kit (Tektrotyd^®^, POLATOM, Otwock, Poland) in accordance with the manufacturer’s instructions. Briefly, freshly eluted ^99m^TcO_4_^−^ (740 MBq) in a 0.9% NaCl solution (pH 7) was added to the vial containing HYNIC-Tyr3-Octreotide (20 µg), tricine and EDDA, mixed and incubated at 80 °C for 30 min according to existing recommendations [23].

#### 2.1.3. Imaging Procedures

##### ^99m^Tc-HYNIC-TOC Scintigraphy

Static planar spot view images of the whole body were performed to evaluate the involvement of salivary glands and major and minor joints. Each spot view image was acquired for 10 min starting 3 h after intravenous (i.v.) injection of ^99m^Tc-HYNIC-TOC (approximately 370 MBq) using a 512 × 512 matrix. The day before the study, the patients were given oral Lugol solution to prevent the uptake in salivary glands of free ^99m^TcO_4_^−^ possibly released by the catabolism of ^99m^Tc-HYNIC-TOC. The acquisition protocol used was exactly the same as described earlier [22].

##### ^99m^TcO_4_^−^ Sialoscintigraphy 

Functional salivary gland scintigraphy (sialoscintigraphy) was performed by i.v. injection of ^99m^TcO_4_^−^ (185 MBq). Images were acquired dynamically over the course of 30 min using a 128 × 128 matrix. Lemon juice (2 mL) was given 10 min afterwards in order to stimulate salivary excretion. Time activity curves were generated to evaluate the uptake and secretion patterns. 

All scintigraphic images were acquired with a double-headed gamma camera (Infinia, General Electric, Milwaukee, WI, USA) equipped with a low energy, high-resolution collimator in accordance with a previously described protocol [22].

#### 2.1.4. Image Analysis

The ^99m^Tc-HYNIC-TOC images were analyzed by two observers (KA and JR) independently of each other and blinded to clinical details. Analysis was performed by using a method proposed before [21]. Briefly, for semi-quantitative analysis, the calf uptake was used as the reference background. A small region of interest (ROI) was delineated and duplicated for each salivary gland and joint bilaterally: shoulders, elbows, wrists, metacarpophalangeal joints, inter-phalangeal joints, knees, and ankles. We did not consider the hips because of possible artifacts due to high bladder activity. By using the average counts in the ROIs, we calculated a ratio to compare and analyze the findings, according to this formula: Average counts in the ROI in each part of interest/Average counts in the calf, leading to a target-to-background ratio (TBR).

Regarding the functional salivary gland scintigraphy, the following parameters were evaluated: (1) uptake score = target background ratio (TBR) of sum of activity at 6–10 min in both parotid glands before lemon juice administration (temporal region was taken as background) and at 24–28 min, after lemon juice administration; and (2) functional score = ratio between the uptake score before and after administration of lemon juice according to Schall et al. [24].

#### 2.1.5. Statistical Analysis

Full descriptive analysis of the variables of interest was performed in order to comply with the objectives of the study. Frequencies (absolute and relative) and percentages were calculated for the qualitative variables and measures of the central tendency (mean and median) and dispersion (standard deviation and interquartile range) together with the maximum and minimum values for the quantitative variables. A stratified analysis of the TBRs in the control group and also in the patients with SS was performed. The Mann–Whitney test was used to evaluate the differences in the TBRs values in the salivary glands between control patients and SS population. A post hoc analysis revealed a power of 0.85 with the sample that was used. The analysis was performed with the Stata 14.2 SE program.

## 3. Results

### 3.1. Patient Groups

Data was gathered from 62 patients with confirmed primary SS and 30 healthy control subjects. Demographic characteristics of the population and the frequency of symptoms are summarized in Table 1. It is remarkable how much the frequency of the symptoms and the positive laboratory tests results that belong to AECG criteria vary. The salivary gland histopathology, the anti-SSa, and anti-Ro were present in 77%, 62%, and 62%, respectively. Moreover, the high frequency of joint pain (87%) as part of the symptoms which are not included as diagnostic criteria in AECG, is remarkable. The ^99m^Tc-sialoscintigraphy was positive in 27% of patients. 

### 3.2. Salivary Gland Uptake of ^99m^Tc-HYNIC-TOC

The analysis of the TBR obtained for the salivary glands for each group of patients showed higher values in submandibular glands in patients with primary SS with a median of 2.73 and a maximum of 5.11. The median for control patients was 1.09. Regarding the parotid glands, the median and maximum recorded values of TBR were 1.72 and 2.3, respectively. A significant higher TBR of the salivary glands was found in patients with primary SS compared to the control group (*p* < 0.001, Mann–Whitney test). The sensitivity/specificity of the ROC curve for the TBR in the submandibular glands was 0.95 (CI: 0.91–0.98); we found that a TBR >1.18 in submandibular glands correctly identified 92% of the patients with primary SS. 

Figure 1 highlights the differences regarding the median values of TBRs for salivary glands between the control group and the primary SS group.

### 3.3. Comparison between ^99m^Tc-HYNIC-TOC in Salivary Glands and ^99m^Tc-Sialoscintigraphy

When we analyzed the uptake of ^99m^TcO_4_^−^ in sialoscintigraphy we found that only 17 patients with primary SS (27%) showed abnormal findings vs. 60 patients (97%) who visually showed any grade of uptake with ^99m^TcHYNIC-TOC. With semi-quantitative analysis, using a TBR >1.18 as cut-off point, we found that 57 patients (92%) were identified correctly by ^99m^Tc-HYNIC-TOC. 

### 3.4. Joint Uptake of ^99m^Tc-HYNIC-TOC 

Table 2 shows a descriptive analysis of the TBRs for every single joint in both groups. 

In patients with SS the highest values were found in the carpus, followed by the knees while the lowest values were recorded in the distal interphalangeal joints (Figure 2).

Figure 3 shows an example of two patients with SS disease with different degrees of uptake of ^99m^Tc-HYNIC in salivary glands (arrows), carpus, and knees.

## 4. Discussion

This study evaluated, for the first time, the distribution of ^99m^Tc-HYNIC-TOC in salivary glands and joints in a population of patients with untreated primary SS. Since SS is an immunological disorder that may involve also involve other organs and tissues besides the salivary glands, this whole-body imaging technique was able to identify the involvement of the joints. For salivary glands, this imaging modality was found to be better than sialoscintigraphy, which though is part of the AECG criteria to diagnose PSS, nowadays it tends not to be considered as part of the diagnostic criteria. ^99m^Tc-HYNIC-TOC could potentially replace sialoscintigraphy in the diagnostic and prognostic criteria for evaluation of this disease. 

In our population the most prevalent symptoms were keratoconjunctivitis, xerostomia, and joint pain, which are findings frequently reported in the literature as part of a broad variety of clinical manifestations and biological abnormalities. Moreover, it is well known that this variety in symptoms accounts for the delay in diagnosis [25]. The frequencies of symptoms and signs observed in our population with respect to the AECG criteria confirmed the importance of combining them for early diagnosis. Likewise, the observed incidence of positive studies in conventional sialoscintigraphy in our population (27%) was low in comparison to the literature. This may be caused by the fact that the distinction between normal results and minor dysfunction is not always easy to detect and mild glandular impairment and borderline results may be misclassified by subjective judgment. In addition, it is possible that when diagnosis is performed during the early stages of the disease, large functional compromise of the gland may not yet be present. Therefore, the pathophysiological process which is mediating the disease cannot be accurately evaluated with conventional sialoscintigraphy. 

To evaluate the pattern of ^99m^Tc-HYNIC-TOC uptake, a semi-quantitative analysis was performed with the help of TBR, using the calf as background area. Values were higher in the submandibular glands than in the parotid glands, thus showing a more severe involvement of the submandibular glands in our population. Surprisingly, when we compared the TBR of the submandibular glands between primary SS patients and healthy controls, medians were close to 1 for the healthy control patients and above 2.5 for the SS patients with statistically significant difference (*p* < 0.01). The ROC analysis showed that a threshold of 1.18 allowed for correct identification in 92% of the patients. Only two patients showed no uptake of ^99m^Tc-HYNIC-TOC; these patients had serological and ocular tests positive for SS with sicca symptoms. Presumably, the disease had not yet affected the salivary glands but activated immune mechanisms were present. Given the mechanism of action of this radiopharmaceutical our results show its ability to detect the presence of somatostatin-mediated immune cell activation in salivary glands in patients with primary SS. The theoretical model reported in the literature supports the usefulness of this technique: abnormal antibodies and T cell responses to muscarinic type 3 receptors (M3R) have been conceived to be pathogenic in primary SS patients [26]. The presence of autoantibodies against M3R has been reported, suggesting that an immune reaction to M3R reactive T cells was detected in the generation of SS [27]. M3R-reactive T cells have been detected in 40% of patients with SS, suggesting that the M3R immune response in SS might function as an autoantigen recognized by autoreactive T cells [28]. Patients with SS have long been thought to suffer from a M3R-reactive T cell type Th1 condition, which has been supported by high levels of IFN-gamma in the serum and a predominance of Th1 over Th2 cells in the blood. Despite a number of caveats, SS is currently conceived as a model for B cell-induced autoimmune disease [29]. It is known how the activities of these cells are orchestrated by soluble factors of the TNF family, most notably the B cell-activating factor (BAFF) described in the late 1990s [30]. This immunologic setting could explain the ability of ^99m^Tc-HYNIC-TOC to detect the inflammatory process, but also the lack of sensitivity of sialoscintigraphy to detect the disease in SS. Although further experimental evidence is needed to confirm this hypotheses, our results suggest that positivity to ^99m^Tc-HYNIC-TOC may forego a reduction of salivary function. Analyses of gene expression profiles of salivary gland tissue from SS patients have confirmed the presence of chronic inflammation [31] and in vitro analysis suggests that cytokines such as interleukin-1-alpha may affect the process of saliva secretion by inhibiting the release of acetylcholine from cholinergic nerves [32]. In this pathophysiological setting of the syndrome, where an immune-mediated inflammatory process exists, the lack of correlation between the two radiopharmaceuticals is not surprising. While ^99m^Tc-HYNIC-TOC binds to activated lymphocytes and provides information about disease activity, ^99m^TcO_4_^−^ evaluates the functional impairment of glandular parenchyma. Furthermore, although noninvasive imaging techniques such as ultrasonography (US), CT, and MR [33] are being studied and might prove useful in the evaluation of the oral involvement in SS, US was demonstrated to have a high diagnostic accuracy in identifying structural changes in salivary glands [34] .We believe that ^99m^Tc-HYNIC-TOC scintigraphy could have an added value since it is capable to demonstrate active inflammatory processes in salivary glands secondary to SS, and could potentially be used for therapy follow-up as well. 

The evaluation of the ratios at the level of the joints in SS patients showed that higher median values were recorded in the carpus and knees. When the medians for the different joints of the SS patients and healthy subjects were compared, a significant difference between the two groups was found, with healthy patients having medians close to 1 and SS patients above 2 in knees and carpus. This finding could also support the theory that somatostatin receptors can be overexpressed in the active phases of the diseases characterized by endothelial activation and lymphocyte infiltration of the synovial cells [35,36]. Moreover, we already demonstrated in a pilot study in a RA population how all patients who showed positive findings in joints at the baseline ^99m^Tc-HYNIC-TOC scan improved after infliximab therapy [22]. We have also suggested how this molecule is able to identify patients with active disease responding to anti TNF-α therapy. We could not establish a cut-off in the ratios to describe abnormality in the joints. ^99m^TcHYNIC-TOC can identify inflamed joints in patients with SS since this is a disease in which the same immunological process at the level of the salivary glands can also be observed around the epithelial structures of other organs including the liver, kidneys, and lungs. Moreover, half of the patients develop extra-glandular complications including arthritis, interstitial lung disease, nervous system involvement, or tubular nephropathy [37].

We found as limitation a source of confusion bias because we did not control the confounders for the control group (age, gender among others). The analysis of the salivary glands using the planar technique seems to underestimate their appearance and, therefore, SPECT/CT should be the technique of choice for future studies.

One of the strongest points of our study is that our results were obtained by highly trained rheumatologist and nuclear medicine physicians who used highly controlled and standardized methods to evaluate the patients and to decrease the index test bias. We included a representative sample size and used a strict type of evaluation to determine the true-positive SS patients.

As we know that PET tracers have several advantages over SPECT tracers (better resolution, possibility for absolute quantification) we believe that ^68^Ga-labelled somatostatin receptor scintigraphy will eventually replace the radiopharmaceutical we used in this study. Nevertheless, our findings are still important because the behavior of the ^68^Ga-labelled compound will be the same as the ^99m^Tc-labelled radiopharmaceutical. 

## 5. Conclusions

In conclusion, we showed that ^99m^Tc-HYNIC-TOC is an important imaging technique to study SS patients, since it allows us to identify active inflammatory processes not only in the salivary glands, but also in the joints. More studies are required in order to include this imaging modality as part of the diagnostic workout of patients with suspected SS to better define disease activity and the extent of extra-glandular inflammation. Moreover, it may provide an objective parameter to evaluate the response to treatment.

## Figures and Tables

**Figure 1 jcm-08-00763-f001:**
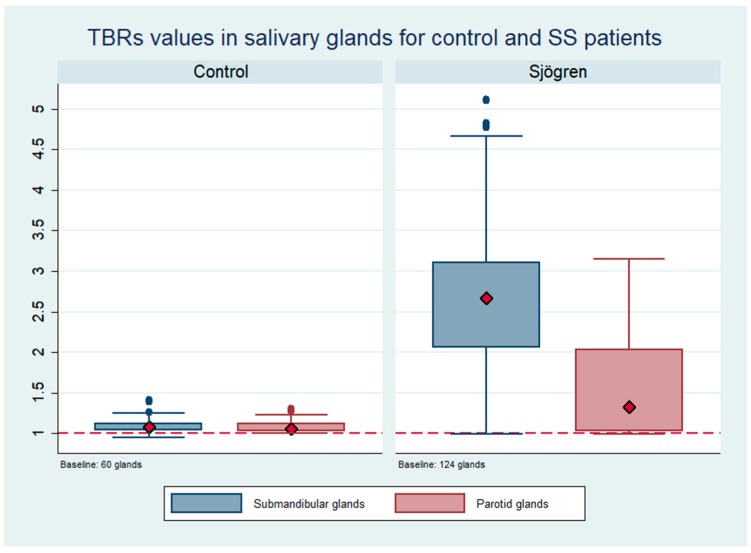
Target background ratio (TBR) in salivary glands of primary SS patients and control subjects.

**Figure 2 jcm-08-00763-f002:**
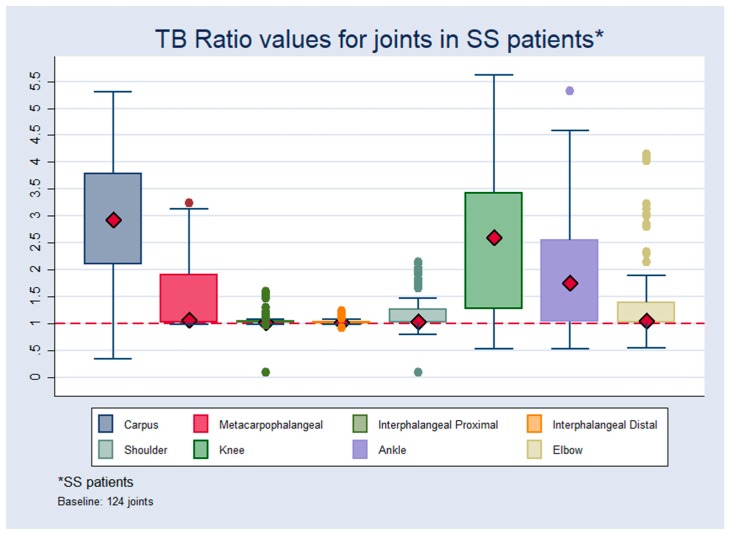
TBR values for joints in patients with SS.

**Figure 3 jcm-08-00763-f003:**
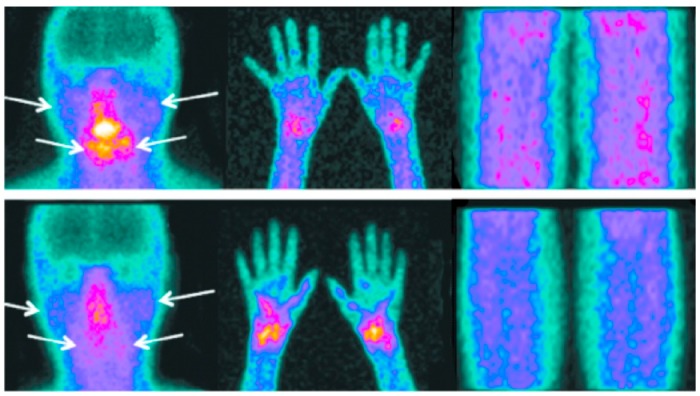
Example of two scans in two patients showing different degrees of ^99m^Tc-SST uptake in the parotid and submandibular glands (arrows), carpus (both patients positive), and knees (both patients negative).

**Table 1 jcm-08-00763-t001:** Demographic characteristics for the total population.

	Control Patients *n* = 30		SS Patients *n* = 62	
	*n*	%	*n*	%
Gender, female	22	73.3	52	83.87
Age, median (range)	58.5	64 (16–80)	48.5	48 (15–71)
12–18 years	1	3.33	1	1.61
19–40 years	1	3.33	15	24.19
41–60 years	9	30.00	34	54.84
>60 years	19	63.34	12	19.35
Dry eye *	-	-	60	96.77
Dry mouth *	-	-	60	96.77
Schirmer test *	-	-	49	79.03
Msg histopathology *	-	-	47	75.80
Sialoscintigraphy *	-	-	17	27.42
Joint pain	-	-	54	87.10

American European Consensus Group (AECG) criteria *. Abbreviations—Msg histopathology: minor salivary gland histopathology; SS: Sjögren syndrome.

**Table 2 jcm-08-00763-t002:** TBR descriptions for joints in SS and control patients.

	TBR Values for Joints in Control Patients					TBR Values for Joints in SS Patients				
	*n*	Median	SD	Min	Max	*n*	Median	SD	Min	Max
**Carpus**	60	1.02	0.16	0.30	1.70	124	2.92	1.09	0.33	5.50
**Metcp**	60	1.10	0.13	0.13	1.70	124	1.06	0.57	0.90	3.20
**Intphpr**	60	1.00	0.13	0.98	1.70	124	1.02	0.14	0.90	1.60
**Intphd**	60	1.00	0.13	0.96	1.70	124	1.02	0.10	0.90	1.20
**Shoul**	60	1.25	0.25	0.95	1.90	124	1.03	0.40	0.90	2.10
**Knee**	60	1.10	0.15	0.98	1.70	124	2.60	1.30	0.90	5.60
**Ankle**	60	1.02	015	0.97	1.70	124	1.00	1.00	0.90	5.30
**Elbow**	60	1.02	0.15	0.94	1.70	124	1.10	0.70	0.90	4.10

Abbreviations—Metcp: metacarpophalangeal; Intphpr: proximal interphalangeal joint; Intphd: distal interphalangeal joint; Shoul: shoulder.

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
