# Peer review of "Value of Somatostatin Receptor Scintigraphy with 99mTc-HYNIC-TOC in Patients with Primary Sjögren Syndrome"

_jcm, 2019, doi:10.3390/jcm8060763_

Reviewer 1 Report

to be checked in the file, mostly spelling and phrasing mistakes

Author Response

Answers to Reviewer 1

We thank the reviewer for taking his time to correct all our spelling mistakes. These have been included in the new version of the manuscript. 

No major criticisms were reported.

 Reviewer 2 Report

The authors evaluated the distribution of 99mTc-HYNIC-TOC in salivary glands and joints of patients with untreated Sjögren syndrome. They also compared this technique with conventional sialoscintigraphy and found it to be more sensitive. Thus, this technique has the potential to replace conventional sialoscintigraphy in diagnostic and prognostic criteria for evaluation of Sjögren syndrome.

Following are my comments:

Line 29………..expand TBR to target-to-background ratio.

Line 32……………….change “significant” to “significantly”.

Line 36…………”maybe” can be changed to “might”

Line 57……………………change “decided join” to “decided to join”

The authors used 30 patients with neuroendocrine tumors as positive control. Did they have a negative control for the study.

Page 3, Materials & Methods………..In the Imaging procedures, the authors reported that images were acquired for 10 minutes starting 3 hours after iv injection for Tc-HYNIC-TOC scintigraphy, whereas, images were acquired over course of 30 minutes for TcO4-Sialoscintigraphy. Is this difference because of difference in techniques? Please explain.

Page 4, Table 1…………….correct the spelling of “Sjogren”, and “Schirmer”

Table 1……last row. Change join to joint

Page 6, when the authors are denoting target-to-background ratio as TBR, it is better to use it consistently throughout the manuscript. Figure 2 title (in the chart) can be written as “TBR values for joints in Sjogren patients”

Author Response

Answers to Reviewer 2

“The authors used 30 patients with neuroendocrine tumors as positive control. Did they have a negative control for the study”.

We apologize for the misunderstanding but indeed the patients with neuroendocrine tumors are our “negative control” and not a positive control group. This is now very clear in the new version of the manuscript.

 Materials & Methods………..In the Imaging procedures, the authors reported that images were acquired for 10 minutes starting 3 hours after iv injection for Tc-HYNIC-TOC scintigraphy, whereas, images were acquired over course of 30 minutes for TcO4-Sialoscintigraphy. Is this difference because of difference in techniques? Please explain.

We used two different scintigraphic techniques for salivary glands. These require different image acquisition protocols. The 99mTc-HYNIC-TOC images are acquired 3 hours after the injection of the radiotracer, with 10 minutes per image. The 99mTc-Sialoscintigraphy images are acquired immediately after tracer injection and last for 30 minutes.

“when the authors are denoting target-to-background ratio as TBR, it is better to use it consistently throughout the manuscript.”

As suggested by the reviewer we checked that all “target-to-background ratio” have been abbreviated with TBR.